# Nucleotide Composition of Ultra-Conserved Elements Shows Excess of GpC and Depletion of GG and CC Dinucleotides

**DOI:** 10.3390/genes13112053

**Published:** 2022-11-07

**Authors:** Larisa Fedorova, Oleh A. Mulyar, Jan Lim, Alexei Fedorov

**Affiliations:** 1CRI Genetics LLC, Santa Monica, CA 90404, USA; 2Department of Medicine, University of Toledo, Toledo, OH 43606, USA

**Keywords:** computational biology, polymorphism, genetic variation, evolution, DNA structure, bioinformatics, genomics

## Abstract

The public UCNEbase database, comprising 4273 human ultra-conserved noncoding elements (UCNEs), was thoroughly investigated with the aim to find any nucleotide signals or motifs that have made these DNA sequences practically unchanged over three hundred million years of evolution. Each UCNE comprises over 200 nucleotides and has at least 95% identity between humans and chickens. A total of 31,046 SNPs were found within the UCNE database. We demonstrated that every human has over 300 mutations within 4273 UCNEs. No association of UCNEs with non-coding RNAs, nor preference of a particular meiotic recombination rate within them were found. No sequence motifs associated with UCNEs nor their flanking regions have been found. However, we demonstrated that UCNEs have strong nucleotide and dinucleotide sequence abnormalities compared to genome averages. Specifically, UCNEs are depleted for CC and GG dinucleotides, while GC dinucleotides are in excess of 28%. Importantly, GC dinucleotides have extraordinarily strong stacking free-energy inside the DNA helix and unique resistance to dissociation. Based on the adjacent nucleotide stacking abnormalities within UCNEs, we conjecture that peculiarities in dinucleotide distribution within UCNEs may create unique 3D conformation and specificity to bind proteins. We also discuss the strange dynamics of multiple SNPs inside UCNEs and reasons why these sequences are extraordinarily conserved.

## 1. Introduction

Over 20 years have passed since the description of ultra-conserved noncoding elements (UCNEs) in mammalian genomes [1]. These numerous and lengthy DNA sequences have been preserved, practically unchanged, for hundreds of millions of years in vertebrates. Their existence and possible roles remain a great enigma in the field of genomics. There are many papers with brilliant descriptions of UCNEs and their astonishing features, among which we could name a few here [2,3,4,5,6,7]. The number of UCNEs in the genome depends on several variable criteria for their definition. In this paper, we bioinformatically studied a public database of 4273 human UCNEs, which have been described by the two criteria: (1) length must be >200 bp and (2) percentage of sequence identity between human and chicken orthologs is ≥95% [4].

In this section, we would like to emphasize specific characteristics of UCNEs that are not the focus of many publications. Since our lab has studied the FTO gene for several years, Appendix A presents an example of ten UCNEs inside the human FTO gene. All ten UCNEs are located inside extra-long introns of the FTO gene. Appendix A demonstrates that BLAST pairwise human–chicken alignments of these UCNEs are much more stringent than the alignment of the coding sequences of human and chicken FTO genes, which harbor these ultra-conserved elements. Note that strong evolutionary conservation of nucleotides remains upstream and downstream of UCNEs (at least 20–50 bp on both sides). Therefore, the borders of UCNEs are rather artificial and determined by a computational algorithm that marks the DNA sequence by an identity threshold of 95%. Since UCNEs are identified through nucleotide sequence identity between species from distant phyla (mammals and birds), UCNEs do not contain DNA-repetitive elements at all, except small simple repeats like short A- or T-runs (e.g., TTTTTT). Thus, UCNEs are unique genomic sequences or exist in only a few copies. As stated by Dimitrieva and Bucher [4] and Habic and co-authors [5], among others, the majority of UCNEs do not share any sequence similarity with other members of ultra-conserved elements. Due to this reason, no sequence motifs have ever been characterized among UCNEs that specify these genomic elements. However, logic tells us that some mysterious biological markers should exist that point to these DNA fragments, making them unchangeable over 300 million years. The first goal of this bioinformatic project was to find any biomarkers that distinguish UCNEs from other genomic fragments. The second goal was to understand the very strange mutational dynamics inside UCNEs. Indeed, in the human genome, there are no “cold spots” for mutations. Hundreds of millions of SNPs are distributed almost randomly over the genome. No lengthy DNA fragment can escape mutations inside it, and the UCNEs are no exception from this rule. Despite numerous SNPs inside UCNEs, only a very limited number of mutations have been associated with human disorders or biological conditions (for details see Habic et al., 2019 [5] and Leypold and Speicher 2021 [6]). Independently, Snetkova et al. (2022) concluded that “there has been no direct demonstration that loss of any ultraconserved enhancer results in reduced viability, fertility, or fecundity” [7]. It is an intriguing mystery as to how numerous mutations inside UCNEs have escaped fixation. UCNEs are far too lengthy to be protein binding sites. Also, they do not fit the modern view of non-coding RNAs, which primarily have evolutionary conservation in structure and not in sequence. Therefore, the hypothesis that there is “fierce purifying selection upon fixation” inside UCNEs is rational yet inexplicable [5,8].

As a result of our computer analysis, we found a unique quality of UCNEs in their dinucleotide composition. This feature should have an influence on the 3D structure of UCNE DNA duplexes. We conjecture that peculiarities in the dinucleotide distribution of UCNEs might create their biological functions through DNA conformation and make them evolutionarily conserved elements.

## 2. Materials and Methods

### 2.1. Databases

The human UCNE database (https://ccg.epfl.ch/UCNEbase/ [4] was downloaded on 2 May 2022 from https://ccg.epfl.ch/UCNEbase/download.php accessed on 4 November 2022, as the text files hg19_UCNEs.fasta.txt and hg19_UCNE_coord.bed. This database was sorted by the UCNEs’ physical order of the chromosomes using our Perl program UCNEprog1.pl. This database contains only two UCNE sequences from the Y-chromosome. Since the 1000 Genomes database does not contain appropriate VCF files for the Y-chromosome, these two Y-chromosome UCNEs have not been processed for SNP distribution. This removal of two Y-chromosome UCNEs from consideration is stated in Section 3, where we specify that 4271 UCNEs (not 4273) were processed.

The 1000 Genomes Project (phase III) [9], which included 2504 individuals from 26 populations, was downloaded in VCF format from the link: (Ftp://Ftp.1000genomes.Ebi.Ac.Uk/Vol1/Ftp/Release/20130502 accessed on 4 November 2022).

Genetic Map tables were downloaded as a Hapmap II combined map (build 37) from ftp://ftp.ncbi.nlm.nih.gov/hapmap/recombination/2011-01_phaseII_B37/genetic_map_HapMapII_GRCh37.tar.gz [10] accessed on 4 November 2022.

Non-coding RNA databases were the following: NONCODE v6 database (http://www.noncode.org accessed on 4 November 2022) of human 173,112 non-coding transcripts was downloaded from the original web site: http://www.noncode.org/download.php [11] accessed on 4 November 2022.

Human database containing 15,056 long lncRNAs (release 2017) was downloaded from UCSC Genome Browser from the link: https://hgdownload.soe.ucsc.edu/downloads.html [12] accessed on 4 November 2022.

### 2.2. Programs for SNP Computational Processing

A subset of the 1000 Genome SNPs within the 4271 UCNE database was generated by our Perl program UCNEsnps.pl, which created 23 VCF files SNPsUCNEvcf2_$chr, (where $chr is 1, 2 … 22, or X). These files are available in an archived compressed form as Appendix A. The alternative allele frequency for each SNP was obtained through our Perl program *1000GfreqSNPsUCNE.pl*, which processed the 8th column of the VCF file, field ‘AF=’, which shows the alternative allele frequency. Additionally, the program *1000GfreqSNPsUCNE.pl* plots the distribution of SNPs into bins by their alternative allele frequency. The number of alternative alleles inside UCNEs for 2504 individuals has been calculated by the program *UCNEsnpINDapr18.pl*, which also creates a table of alternative allele SNP distributions in different regions. The distribution of meiotic recombination rates inside the UCNEs was created by the *GeneticMap_AF.pl* program. The random expectation model for the evaluation of meiotic recombination rates inside randomly distributed 300 bp sequences along chromosomes was performed by our Perl program *RandomPositionsForRecombination.pl*, which creates random positions for so-called “random-UCNEs”. The distribution of meiotic recombination rates inside “random-UCNEs” was calculated by a slightly modified program *GeneticMap_AFrand.pl*. Monte Carlo simulations has been done by multiple execution of *GeneticMap_AFrand.pl*. The oligonucleotide composition of UCNEs and all human chromosomes was calculated using our previously published program *NTcomposition.pl* [13]. The genomic signature (ρ) was calculated based on the relative frequencies of nucleotides and dinucleotides, following the formula by Karlin and Burge [14]:$$\rho_{xy}=\frac{F_{xy}}{F_{x}\times F_{y}},$$
where ***F_xy_*** is a relative frequency of dinucleotides ***xy*** among all 16 possible dinucleotides; and ***F_x_***, ***F_y_*** are relative frequencies of nucleotides ***x*** and ***y*** among all four possible nucleotides A, G, C, T.

BLAST results of UCNE sequences against ncRNA DBs was calculated by a local BLAST program installed from the latest version of NCBI (May 2022) using the following command line: “blastn -query UCNEfasta_sorted.txt -db lncRNA.fa -evalue 0.0001 -num_alignments 1 -out blast_UCSC”. We used the single best-match output option and a low threshold for alignment similarity (*p*-value cutoff of 0.0001). All Perl programs are available on our website (http://bpg.utoledo.edu/~afedorov/lab/UCNE.html accessed on 4 November 2022) in a package that includes an Instruction Manual (UCNEinstruction.docx) and Protocols (UCNEprotocols.docx). In addition, this package of programs, instructions, and protocols is available in the Appendix A.

### 2.3. Statistics

Standard error for the genomic signatures of UCNE sequences was calculated via re-sampling statistics (bootstrap approach). We obtained 1000 random subsets from 4273 UCNE sequences, each containing 50% (2137) of the entire sample. For each random subset, the genomic signatures were calculated. Finally, from variations in this 1000 subset distribution, standard error was calculated. Bootstrap calculations have been performed using our Perl pipeline programs (BootstrapUCNE.pl; NTcomp.pl; startNTcompRAND.pl; and GenomicSignature.pl), which are available from the Appendix A, together with protocols.

## 3. Results

### 3.1. Database

Among several public human UCNE databases, we chose the one created by Dimitrieva and Bucher [4] because it is one of the oldest datasets, is brilliantly described in the paper, and has an interactive website and smart identifiers for each UCNE element. This database was downloaded and all sequences and identifiers were arranged strictly by the UCNEs’ physical order on the chromosomes. A total of 56 sequences (1.3%) were removed due to inconsistencies with their identifiers. These reorganized files, named UCEfasta_sorted.txt for 4273 sequences and UCEids.txt for UCNE identifiers and positions, are available from the Appendix A.

### 3.2. Density of SNPs inside UCNE vs. Whole Genome

The 1000 Genomes Database, version phase 3, contains 81,042,272 SNPs representing point mutations inside 2,867,437,753 bp of the sequenced human genome (version Build 37). Of these human SNPs, we computationally filtered 31,046 SNPs located inside 4271 human UCNEs, the total length of which is 1,393,448 bp. Therefore, the density of SNPs inside UCNE sequences (22 SNPs per 1000 nucleotides) is only 24% less than in the whole genome (28 SNPs per 1000 nucleotides). A vast majority of human SNPs represent rare alleles, for which the alternative allele has a frequency of less than 1% across all populations. The distribution of the number of human SNPs and their subset inside UCNEs by their alternative allele frequencies, are shown in Table 1. In this table, SNPs were divided into 100 bins based on their alternative allele frequency. Every bin had the same size of 1%. The first bin contains all SNPs with alternative allele frequencies from 0 to 1%, the second bin contains SNPs with alternative allele frequencies between 1% and 2%, and so on. Since the size of the whole genome is 2000 times larger than the size of our UCNE sequences, the number of SNPs in the whole genome is much larger than for SNPs inside UCNEs. Therefore, in order to compare the distribution of SNPs inside UCNEs versus the whole genome, we calculated the relative frequencies of these SNPs in the bins by dividing their number inside the bin by the total number of SNPs for the entire dataset. These relative frequencies are shown in columns 3 and 5 of Table 1, while their distribution is shown in Figure 1. Since the relative frequency of the SNPs in the first bin (0–1%) is more than 20 times larger than the rest of the bins, the first bin was excluded from Figure 1 in order to remove scale distortion. Figure 1 also does not show the last 50 bins (from 51% to 100%) because these frequent alternative alleles are highly enriched by derived alleles instead of ancestral ones, which causes wrong conclusions (e.g., see Paudel and co-authors [15]). However, the full set of data for all 100 bins are presented in the Appendix A. The relative frequency of rare allele SNPs in the first bin (0–1%) is higher in UCNEs (92.7%) than in the whole genome (84.4%) (see Table 1). For the rest of the bins, the situation is the opposite. Figure 1 demonstrates that the relative frequency of SNPs in 2–50 bins is significantly higher in the whole genome than inside UCNE sequences. Moreover, in the first ten bins (from #2 to #11), the difference in SNP relative frequencies between the whole genome versus UCNEs is, on average, 1.8 times, while in further bins, the differences start increasing, reaching, on average, 3.2 times for the alternative alleles with frequencies in the range of 30–50%. Table 1 and Figure 1 demonstrate that the mutations that occur inside UCNEs are nearly as frequent as in the whole genome, but something prevents their propagation towards fixation. Our data is in good agreement with the papers by Habic et al. (2019) [5] and Katzman et al. [8] and support the idea that there is some unknown process that actively prevents the fixation of mutations inside the UCNE elements (see further discussion in the Section 4).

### 3.3. Number of Mutations inside UCNE among 2504 Individuals

The 2504 human genome sequences from the 1000 Genome database have been analyzed in order to calculate the number of mutations per person inside the UCNE elements. As the vast majority of alternative alleles with low frequencies are derived (mutant) alleles in the 1000 Genomes Database [9], we calculated the number of alternative alleles inside the entire set of 4271 UCNEs in every person, which is practically equivalent to the number of mutations with UCNEs. Because there are well-known problems regarding the misclassification of abundant alternative alleles that may not be derived but are ancestral alleles, we did not compute the SNPs that have alternative alleles with frequencies above 50%. This truncation guarantees that we do not overestimate the number of mutations inside UCNE per person. Our data is presented in Figure 2, while the exhaustive data for every individual is available in the Appendix A. Figure 2 shows that the minimal number of mutations inside UCNEs was 285 in individual ‘NA12400’ from the European CEU population, while the maximal number of mutations was 536 in individual ‘NA18923’ from the African YRI population. The average number of UCNE mutations per person in five regions is shown in Table 2, which represents the described above data for alternative alleles with frequency cutoffs of 50% and additional calculations for exclusively rare alternative alleles with cutoff frequencies of 2% (presented in Appendix A). Table 2 and Figure 2 demonstrate that every person has numerous mutations inside UCNE sequences. African populations have considerably more mutations than the other four regions. Moreover, this excess of UCNE mutations in Africa over the rest of the World predominantly comes from the rare alleles with frequencies less than 2%. By conservative estimation, every person has more than 300 mutations within their UCNE sequences. This colossal number of UCNE mutations per person presents a problem in explaining how it can be possible that these mutations could be removed by natural selection (see further discussion in Section 4).

### 3.4. Non-Coding RNAs inside UCNEs

Could UCNEs represent non-coding RNAs? This question is tricky because the majority of ncRNAs are expressed at extremely low levels [16]. In our previous research of ten intronic UCNEs inside the FTO gene [17], we found several matches of each UCNE with the Sequence Read Archive (SRA) database [18]—The largest repository of human transcripts. However, such low-level hits may be explained by the experimental contamination of RNA sequences by pre-mRNA or DNA molecules. Therefore, no definite conclusions have been drawn. In this paper, we performed an exhaustive pairwise BLAST alignment of 4273 UCNE sequences against (1) 173,112 human ncRNAs from the NONCODE database (total length 290,248 kb) [11]; and (2) 15,056 very long lncRNAs from the UCSC database (total length 516,136 kb) [12]. These BLAST results are presented in the Appendix A. In the first case, only 12.7% of UCNEs showed similarity hits with the NONCODE database, and in the second case, 16.5% of UCNE sequences produced hits with lncRNAs. Moreover, a significant portion of these BLAST hits with NONCODE and lncRNAs are not perfect matches nor represented by small fragments (30–50 bp) of UCNE sequences. Such non-perfect hits may be interpreted as alignments with genomic UCNE duplicates and not genius UCNEs. Essentially, 87.3% of UCNEs do not match with NONCODE, and 83.5% of UCNEs do not match UCSC lncRNAs. Since these ncRNA databases cover 10% (NONCODE) and 18% (lncRNA) of the entire human genome, we concluded that the observed BLAST hits are random matches due to the large total length of investigated ncRNAs. Hence, UCNEs do not represent non-coding RNAs.

### 3.5. Meiotic Recombination Rates inside UCNEs

Recombination rate is a critical parameter for SNP dynamics and probability of propagation of mutations toward fixation [19]. In the human genome, the meiotic recombination rate could differ thousands of times along the chromosome. There are “hot” and “cold” spots for the recombination rate [20]. Using the recombination rate database [10] we computed the recombination rate inside our 4271 UCNE set in order to explore the possible association of UCNE with chromosomal regions of low or high recombination.

This examination demonstrated that many UCNEs have a very low rate, while many others have a very high meiotic recombination rate inside them. At the same time, the average recombination rate inside UCNEs is about the same as in the whole genome. We used Monte Carlo simulations to generate “random-UCNEs” of a 300 bp length that are randomly distributed along chromosomes. Figure 3 demonstrates the distribution of recombination rates inside real UCNEs (red) versus “random-UCNEs” (blue). These two distributions are very similar to each other, with the exception of a small portion (~5%) of “random-UCNEs”, which tend to have very low recombination rates compared to real UCNEs (observe left part of Figure 3, where blue columns are higher than red ones). However, this minor difference could be explained by the fact that some “random UCNE” positions may be located inside non-sequenced genomic gaps of the Build 37 version of the whole genome, which we did not consider. All in all, we did not find any significant preference of real UCNEs to be located within chromosomal regions with a particular meiotic recombination rate.

### 3.6. Search for UCNEs Sequence Markers

Since the discovery of UCNEs twenty years ago, scientists still have not found any clues or biomarkers that would explain why these DNA fragments remain practically unchanged over hundreds of millions of years of evolution. The majority of human UCNEs are unique or present in a few genomic copies [4]. Different UCNEs do not have sequence similarity with each other. Thus, sizable DNA fragments common between UCNEs are clearly absent. UCNE markers may be present at the borders of these elements, such as major transcription factor binding sites located in front of genes. We made several attempts using multiple alignment programs to compare UCNEs with 3 kb 5′- and 3′-flancking regions to find common sequences without any success (our unpublished results). So, sizable sequence motifs (>10 nucleotides) that might mark UCNEs as unchangeable DNA are probably absent. We also searched for possible nucleotide inhomogeneity regions (e.g., H-DNA, Z-DNA, among others) inside UCNEs using our old programs [13,21]. As a result, no significant sequence non-randomness inside UCNEs has been found (our unpublished data). There is a possibility that very short and numbered nucleotide sequences are markers for UCNEs. To explore this hypothesis, we analyzed the possible peculiarities in oligonucleotide distribution inside UCNEs. Using our Perl programs [13], we investigated the oligonucleotide composition of the UCNE database from single nucleotides to 8-mer oligonucleotides. These data for all chromosomes are present in the Appendix A and a fragment of it for 1 to 3-mer oligonucleotides is illustrated in Table 3. Several peculiarities in UCNE nucleotide composition have been found. Firstly, UCNEs are, in general, C + G poor sequences. The average C + G composition in UCNEs is 36.8%, in contrast to the genome average of 41%. Secondly, UCNEs do not include CpG-islands, so the frequency of CpG dinucleotides inside them is about the same as in the whole genome on average (the UCNE genomic signature is ρ_CG_ = 0.27; while the genome average is ρ_CG_ ~ 0.24, Table 4). Thirdly, UCNEs are 28% enriched by GpC dinucleotides at the expense of GG and CC dinucleotides (from here, we used the notation GpC of adjacent nucleotides on the same strand to distinguish them from G–C Watson–Crick pairs). We also found that a group of longer oligonucleotides are overabundant, while another group of oligonucleotides are underabundant inside UCNEs. For example, among 4-mer sequences with a balanced 50%-CG composition, TGCA, AGCA, TGCT, ACAG, and TCTG are the most overrepresented ones (Appendix A). Yet, these 4-mers are not present in special alignments inside UCNE and not in every UCNE. So, it is unlikely that they are the sole markers for ultra-conserved DNA. Among the longest studied oligonucleotides represented by 8-mer sequences, obviously, the most overabundant are AT-rich ones, such as AAAAAAAA, because UCNEs are GC-poor sequences. Among 8-mers with rich GC-composition, AGCAGCAG, CAGCTGCT, CAGCTGTG, and CAGCTGCA are the most overrepresented, as shown in the Appendix A. However, each of these overrepresented 8-mers are only present inside a minor fraction of UCNEs and their positions and alignments to each other are random. Presumably, larger oligomers could not be UCNE important markers. For this reason, we focused our examination on the non-randomness of the dinucleotide distribution inside UCNEs.

One of the most important parameters in the investigation of dinucleotide occurrences is the so-called genomic signature (ρ_XY_) introduced by Karlin and Burge, which measures the preferences of two nucleotides X and Y to form a dinucleotide XY on the same DNA strand [14]. When ρ_XY_ = 1, there is no preference for these two X and Y bases to form an XY pair. When ρ_XY_ < 1, these nucleotides avoid the formation of the XY dinucleotide. When ρ_XY_ > 1, there is a non-random predisposition for the X nucleotide to be in front of Y. The more ρ_XY_ deviates from 1, the stronger the non-randomness in the formation of the XY pair. These genomic signatures are unique markers for biological species [14]. Genomic signatures, calculated for the entire UCNE set, as well as the whole human genome, are shown in Table 4. The most significant ρ variations between UCNE and the genome average were observed for the GpC dinucleotide (a 28% increase in ρ value inside UCNEs). The CC and GG dinucleotides experience a 14% decrease inside the UCNE compared to the genome average. Since we processed the entire set of 4273 UCNEs with a total length of 1.4 Mb, these variations are statistically significant. Bootstrap statistical analysis demonstrated that a standard deviation of the GpC genomic signature for UCNEs is 0.004. All in all, these peculiarities in the dinucleotide UCNE compositions may be significant for changes in the DNA double helix structure, which is the focal discussion in Section 4 below.

## 4. Discussion

### 4.1. Strong Nucleotide Stacking Interactions within UCNEs

The three-dimensional structure of the double-stranded DNA helix is formed by two types of nucleotide interactions: (i) Watson–Crick base pairing of nucleotides from opposite strands and (ii) Pi-stacking interactions between adjacent nucleotides from the same strand. On average, the stacking interactions between nucleotides is the major contributing factor to the stability of the DNA duplex, not base pairing [22,23,24,25,26]. Nucleotide stacking interactions are implemented by the third type of Van Der Waals or London dispersion forces, which are perhaps inherently quantum mechanical and still not fully appreciated [27,28]. There are some controversies regarding the measurement of stacking forces in a DNA duplex [29]. Free energies of stacking interactions, measured in various experimental settings of the DNA melting process, unanimously revealed the strongest stability of GpC, followed by CpG, than other dinucleotide combinations [23,30,31,32]. Single molecule mechanical experiments using DNA origami also confirmed the lowest stacking minimum free energy for GpC dinucleotide [33]. Moreover, the GpC dinucleotide dissociation rate is 100 times lower compared to any other combination of adjacent nucleotides (500 s^−1^ versus 50,000 s^−1^) [33]. Independently, theoretical quantum chemical studies of stacking energy in the gas phase model determined the most stable steps, GpC followed by CpG [34,35,36]. Since the GpC dinucleotide is the most overabundant above random expectations inside UCNEs, we hypothesized that the UCNE sequences may form a DNA duplex with distinctive properties. Inside UCNEs, 14% of CC dinucleotides and 14% of GG dinucleotides were replaced by GpC dinucleotides, producing a 28% relative excess of the GpC. For evaluation of the increase in DNA stability of UCNEs, we must know the difference in stacking energy between GpC versus CC and GG dinucleotides. Svozil et al.’s (2010) paper provides stacking energies for all dinucleotide pairs for DNA molecules calculated using gas spectrometry [36]. According to these authors (Table 2 therein), GpC dinucleotide has the strongest stacking energy (−14.14 kcal/mol), while the GG and CC dinucleotides have the weakest stacking (−7.85 kcal/mol) among all possible dinucleotides. Klichher et al.’s (2016) paper [33] also estimated GpC stacking free energy (ΔG = −3.41 kcal/mol) as being twice as strong as GG or CC (ΔG = −1.64 kcal/mol). This is congruent with Yakovchuk et al. (2006) [23], who estimated GpC stacking free energy as ΔG = −2.17 kcal/mol vs. GG and CC stacking as ΔG = −1.44 kcal/mol. However, Santa Lucia (1998) published less dramatic differences between GpC (ΔG = −2.24 kcal/mol) versus GG and CC (ΔG = −1.84 kcal/mol) dinucleotides stacking free energy [37]. All listed publications suggest that UCNE DNA should have very strong duplex structure. Recently Beyerle et al. (2021) demonstrated that a regulatory protein access to the DNA duplex is thermally driven by base stacking–unstacking interactions [38]. Therefore, the distinctive stacking properties of UCNEs should provide peculiarities in their interactions with DNA binding proteins. This conjecture is congruent with the experimental data by McCole et al. (2018), which associated UCNEs at specific places in the three-dimensional mammalian genome organization model [39].

### 4.2. Paradox for Purifying Selection of Numerous Mutations in UCNEs

We demonstrated that every human has more than 300 mutations within the investigated set of 4271 UCNEs (Figure 2). Simple combinatorics suggest that three hundred mutations should, on average, form 5.3 UCNEs, for which both maternal and paternal UCNEs have mutations inside the same UCNE sequence (150 × 150/4273 = 5.3). Therefore, each person should be a compound homozygote for several mutant UCNE sequences and, in addition, be a heterozygote for at least 300 mutations inside UCNEs. Surprisingly, UCNEs have remained practically unchanged for the 300 million years since the last common ancestor between mammals and birds [40]. The computational modeling demonstrated that after a particular threshold of deleterious mutation influx, the purifying selection is unable to keep up with the rate of deleterious mutations, and they start to accumulate to fixation [19]. Hence, it is impossible to select out 300 mutations per individual. Figure 1 and Table 1 show that, while the rare alleles are relatively overabundant inside UCNEs, the number of common SNPs with an alternative allele frequency is 30–50% inside UCNEs and is 3.2 times less than averagely expected for the whole genome. Prevention of fixation of numerous rare UCNE mutations is a paradox, which is currently unexplainable. Below, we propose our two conjectures that may resolve the paradox.

The first idea is based on the notion that the effectiveness of natural selection is in direct proportion to the number of offspring per individual that compete with each other for the survival of the fittest [19]. The natural selection of UCNEs, which is ineffective on a whole human organism due to the limited number of offspring, may still work on the level of single-cell gametes. Since every male produces millions of spermatozoids, the selection against a large number of mutations may be effective at this level. For this scenario, mutations inside UCNEs should be associated with the gamete fitness. A natural competition among millions of spermatozoids should tremendously increase the power of natural selection.

The second conjecture is that natural selection itself is not the major force for UCNE SNP dynamics, but instead some unknown molecular process. For example, there is a significant excess of mutations that convert G–C base pairs into A–T base pairs in the human genome than the reverse; mutations that convert A–T base pairs into G–C. However, there is no deterioration of GC-content in humans because the initial excess of G–C to A–T mutations is compensated by the Biased Gene Conversion that operates on the level of DNA reparation of mismatched base pairs in DNA heteroduplexes [15].

All in all, the paradox of the existence of numerous ultraconserved elements is unresolved and is awaiting discovery by researchers. To finalize our paper, we would like to cite the conclusion of the comprehensive review by Snetkova and co-authors: “Since ultra-conserved constraint is likely to be due to a combination of factors, future work should explore evidence for all potential drivers more fully…” [7].

## 5. Conclusions

UCNE sequences are AT-rich and enriched by GpC dinucleotides;Every human has over 300 mutations inside 4273 UCNE;We hypothesized that due to unique dinucleotide composition UCNE sequences may form a DNA duplex with distinctive properties. This hypothesis is awaiting experimental testing.

## Figures and Tables

**Figure 1 genes-13-02053-f001:**
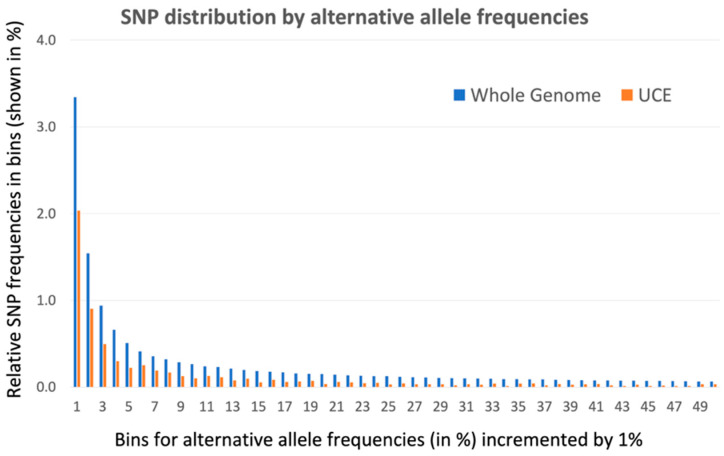
Distribution of SNP relative frequencies by their alternative allele abundance inside UCNEs and the whole genome. This is a graphical representation of data from Table 1 for the second up to fiftieth bins for columns 3 and 5. Starting from the second bin, the relative frequency of SNPs inside the whole genome is always higher than inside UCNE sequences, and the difference becomes more dramatic with the increase of alternative allele frequency (bin consecutive order).

**Figure 2 genes-13-02053-f002:**
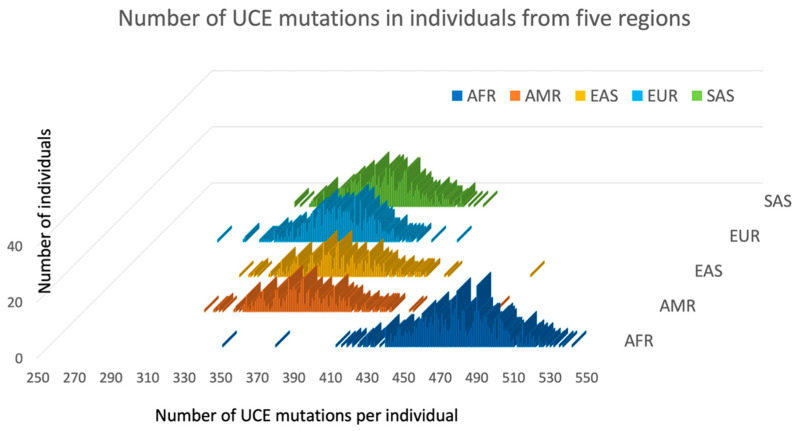
Number of alternative alleles with the frequencies up to 50% inside the 4271 UCNE sequences among 2504 individuals from five regions. Individuals are represented in five groups, depending on their ethnicity and according to their classification in the 1000 Genomes Database. AFR represents African populations (navy blue), AMR—Americans populations (red), EAS—East Asian (yellow), EUR—Europeans (blue), and SAS—South Asia (green). Each individual is represented by a colored bar, and its position along horizontal axis corresponds to the total number of alternative alleles inside the UCNEs in each person.

**Figure 3 genes-13-02053-f003:**
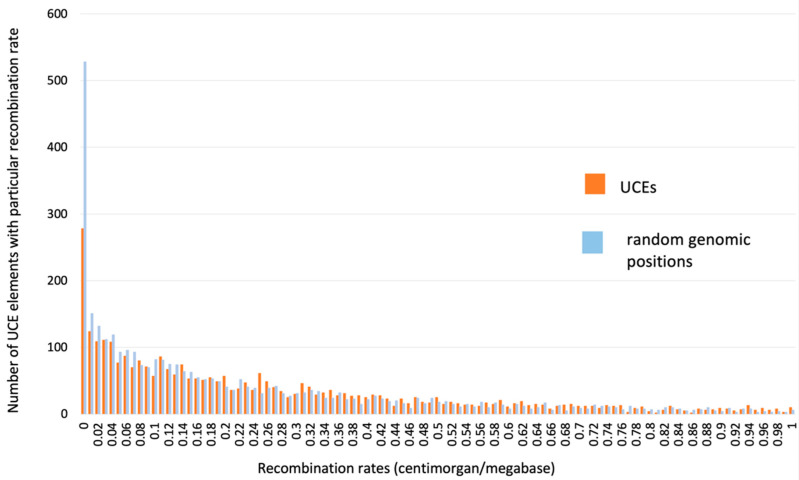
Distribution of meiotic recombination rates inside UCNEs versus random genomic positions (so-called “random-UCNEs”). Recombination rates were divided into equal-sized intervals of 0.02 centimorgans (cM) per one million nucleotides, which are shown along the horizontal axis. The number of UCNE and “random-UCNE” sequences that have a recombination rate within a particular interval (bin) are plotted along the vertical axis.

**Table 1 genes-13-02053-t001:** Distribution of SNPs by their alternative allele frequencies inside UCNEs and the whole genome. SNPs are divided into one hundred bins by their alternative allele frequencies shown in column one. Columns 2 and 4 show the number of SNPs in the corresponding bin inside the whole genome and UCNEs, respectively. Columns 3 and 5 show the relative frequencies of SNPs in the bins by dividing the number of SNPs in the bin by the total number of analyzed SNPs in the whole genome and UCNEs, respectively. The entire table for 100 bins is shown in the Appendix A. The graphic of distribution of relative frequencies of SNPs (columns 3 and 5) inside the bins is illustrated in Figure 1.

Bins for Alternative Allele Frequency	Whole Genome		Ultra Conserved Elements Only	
	Number of SNPs inside Whole Genome	Relative Frequency (%) of SNPs inside Whole Genome	Number of SNPs inside UCEs	Relative Frequency (%) of SNPs inside UCEs
0–1%	68,430,653	84.438	28,787	92.724
1–2%	2,709,034	3.343	632	2.036
2–3%	1,249,017	1.541	280	0.902
3–4%	761,505	0.940	154	0.496
4–5%	536,314	0.662	93	0.300
5–6%	411,115	0.507	69	0.222
6–7%	334,473	0.413	78	0.251
7–8%	287,678	0.355	59	0.190
8–9%	258,931	0.320	52	0.167
9–10%	231,334	0.285	39	0.126
10–11%	213,325	0.263	31	0.100
11–12%	193,665	0.239	40	0.129
…				
0–100%	81,042,272 total	100%	31,046 total	100%

**Table 2 genes-13-02053-t002:** Average number of mutations inside UCNE sequences per person in five regions calculated for two cutoffs (2% and 50%) for alternative allele frequencies.

Region	Average Number of Alternative Alleles Per Person in a Region	
	Cutoff 50%	Cutoff 2%
Africa	472	117
America	370	47
Europe	352	42
East Asia	373	40
South Asia	357	46

**Table 3 genes-13-02053-t003:** Distribution of oligonucleotides inside 4273 UCNE sequences. The relative frequency of an *n*-mer oligonucleotide was calculated by dividing the number of occurrences of this oligonucleotide by the total number of occurrences of all *n*-mer oligonucleotides. The sum of all relative frequencies for all oligonucleotides of the same size is equal to 1. The entire distribution of 1 to 8-mer oligonucleotides for all human chromosomes and UCNEs is shown in the Appendix A.

Oligo-Nucleo-Tides	UCNE Sequences		Chromosome #1		Oligo-Nucleo-Tides	UCNE Sequences		Chromosome #1	
	Relative Freq (%)	Number of Occurrences	Relative Freq (%)	Number of Occurrences		Relative Freq (%)	Number of Occurrences	Relative Freq (%)	Number of Occurrences
**1-mer**					**3-mer**				
A	0.314	445,884	0.291	67,070,277	TTT	0.041	58,385	0.037	8,583,142
T	0.317	449,114	0.292	67,244,164	TTC	0.020	27,875	0.020	4,548,877
C	0.183	260,157	0.208	48,055,043	TTG	0.021	29,834	0.019	4,344,678
G	0.185	262,642	0.209	48,111,528	TCA	0.022	30,484	0.020	4,522,569
**2-mer**					TCT	0.020	27,643	0.022	5,129,424
AA	0.110	156,199	0.095	21,901,540	TCC	0.011	15,703	0.015	3,657,040
AT	0.092	129,314	0.074	17,121,783	TCG	0.002	3345	0.002	535,651
AC	0.048	68,447	0.050	11,598,278	TGA	0.022	30,724	0.019	4,486,632
AG	0.064	90,715	0.071	16,448,644	TGT	0.023	32,190	0.020	4,584,113
TA	0.075	106,391	0.063	14,554,789	TGC	0.017	24,240	0.015	3,357,313
TT	0.112	157,604	0.096	22,048,241	TGG	0.013	18,970	0.019	4,368,306
TC	0.055	77,483	0.060	13,844,699	CAA	0.021	29,151	0.019	4,288,540
TG	0.075	106,452	0.073	16,796,378	CAT	0.021	29,829	0.018	4,120,946
CA	0.074	104,633	0.073	16,768,284	CAC	0.012	17,466	0.015	3,506,405
CT	0.064	90,607	0.071	16,444,797	CAG	0.020	27,845	0.021	4,852,390
CC	0.036	51,183	0.054	12,466,763	CTA	0.012	17,425	0.013	2,941,433
CG	0.009	12,748	0.010	2,375,159	CTT	0.020	28,116	0.020	4,634,644
GA	0.055	77,358	0.060	13,845,615	CTC	0.012	16,714	0.018	4,057,534
GT	0.050	70,436	0.050	11,629,291	CTG	0.020	28,042	0.021	4,811,169
GC	0.044	62,159	0.044	10,145,272	CCA	0.013	18,634	0.019	4,330,820
GG	0.037	51,737	0.054	12,491,312	CCT	0.013	18,288	0.019	4,273,302
**3-mer**					CCC	0.008	11,006	0.014	3,193,020
AAA	0.041	57,532	0.037	8,516,543	CCG	0.002	3016	0.003	669,612
AAT	0.034	48,424	0.024	5,470,905	CGA	0.002	3173	0.002	523,798
AAC	0.015	21,576	0.014	3,332,435	CGT	0.002	3453	0.003	597,422
AAG	0.020	28,302	0.020	4,581,648	CGC	0.002	2967	0.003	579,316
ATA	0.022	30,622	0.019	4,475,100	CGG	0.002	3096	0.003	674,618
ATT	0.034	48,554	0.024	5,500,468	GAA	0.020	27,818	0.020	4,518,460
ATC	0.014	19,839	0.013	3,035,996	GAT	0.014	20,080	0.013	3,056,974
ATG	0.021	30,019	0.018	4,110,209	GAC	0.009	12,409	0.010	2,216,474
ACA	0.022	30,869	0.020	4,553,751	GAG	0.012	16,823	0.018	4,053,693
ACT	0.016	22,142	0.016	3,732,934	GTA	0.012	16,988	0.011	2,566,721
ACC	0.008	11,883	0.012	2,725,309	GTT	0.016	22,177	0.014	3,329,970
ACG	0.002	3335	0.003	586,276	GTC	0.009	12,839	0.010	2,202,280
AGA	0.019	27,353	0.022	5,150,760	GTG	0.013	18,242	0.015	3,530,308
AGT	0.016	22,472	0.016	3,719,675	GCA	0.017	24,365	0.015	3,361,131
AGC	0.016	22,161	0.014	3,317,232	GCT	0.016	22,187	0.014	3,309,131
AGG	0.013	18,369	0.018	4,260,968	GCC	0.009	12,391	0.013	2,891,387
TAA	0.029	41,225	0.020	4,577,976	GCG	0.002	2995	0.003	583,618
TAT	0.022	30,705	0.019	4,472,951	GGA	0.011	15,843	0.016	3,684,403
TAC	0.012	16,780	0.011	2,542,958	GGT	0.009	12,082	0.012	2,728,078
TAG	0.012	17,407	0.013	2,960,898	GGC	0.009	12,562	0.013	2,891,408
TTA	0.029	41,106	0.020	4,571,528	GGG	0.008	11,045	0.014	3,187,415

**Table 4 genes-13-02053-t004:** Genomic signatures (ρ) of the UCNE sequences versus the whole genome. Note that complementary dinucleotides (e.g., TG and CA) have the same genomic signatures. Green color highlights the overabundant dinucleotide GpC, while the red color the underabundant CC and GG dinucleotides. Standard errors are shown for ρ(UCNEs). Different human chromosomes have slightly different nucleotide compositions, as shown in the Appendix A. Therefore, their genomic signatures vary from chromosome to chromosome with fluctuations of about 1%.

Dinucleotide	ρ (Genome)	ρ (UCNEs)
CG	0.24	0.27 ± 0.002
GC	1.02	1.30 ± 0.004
TA	0.74	0.76 ± 0.002
AT	0.88	0.92 ± 0.002
CC/GG	1.24	1.08 ± 0.004
TT/AA	1.12	1.12 ± 0.002
TG/CA	1.20	1.28 ±0.003
AG/CT	1.16	1.10 ± 0.003
AC/GT	0.83	0.84 ± 0.003
GA/TC	0.99	0.94 ± 0.003

## Data Availability

All Appendix A and Perl programs are available on our website (http://bpg.utoledo.edu/~afedorov/lab/UCNE.html accessed on 4 November 2022) in a package that includes an Instruction Manual (UCNEinstruction.docx) and Protocols (UCNEprotocols.docx).

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
