# Peer review of "Nucleotide Composition of Ultra-Conserved Elements Shows Excess of GpC and Depletion of GG and CC Dinucleotides"

_genes, 2022, doi:10.3390/genes13112053_

Round 1

Reviewer 1 Report

Paper is well written and presents interesting results that are important for the whole area of Genomics.  However, there are several things should be done to improve the manuscript prior to publication.

 1) Statistics is acceptable, yet the standard deviation for genomic signatures in the Table 4 (page 12) is not presented neither in the table itself nor in the legend to it.  The description of this standard deviation is hidden in the middle of the text (page 9, line 365) and easily escapes attention of possible readers.  This Table 4 presents the most critical results for the entire paper and, therefore, standard deviation values should be presented explicitly inside this table.

 2) The authors missed very important review on the UCNE, which was written this year in Nat Rev Genet. 2022 March; 23(3): 182–194 (PMID: 34764456) about possible functions of ultra-conserved sequences and the latest achievements in this area.  This paper should be cited  in both introduction and discussion sections.

 3) The authors should also cite the paper published in Science 2007, VOL. 317, NO. 5840 (PMID: 17702936), in several places where they discuss natural selection forces.  This manuscript by Katzman and co-authors is the first publication that thoroughly investigated the selection forces inside UCNE.  These authors clearly demonstrated that ultra-conserved sequences are under negative selection that is much stronger than that in protein coding genes.

 4) English should be improved in several places. For example, there are several language mistakes: 

Page 1, line 22:   instead of “extraordinary”  it should be “extraordinarily”.

Page 3 line 120:  instead of “simulations has been done my multiple execution” it should be “simulations has been done by multiple execution”.

Page 14, line 494: misspelling the word “reseachers”

Finally, I suggest improving the language in a short paragraph on page 6, starting from line 220.

Reviewer 2 Report

This paper has several issues that need addressing.

The title is misleading as the results described in the paper do not address a “unique DNA conformation,” or justify the statement in the Discussion that “UCNE sequences may form a DNA double helix with unique properties.”  Rather, a major contribution of this paper is the observation that UCNEs exhibit an over representation of GC dinucleotides with a corresponding decrease in CC and GG dinucleotides.  From this bioinformatic analysis the authors try to invoke ideas about mutations (mutational dynamics), sequence evolution, and DNA function; ideas that, at times, do not seemingly comport with an appreciation of our current understanding of biology.  Some ideas and arguments in the Discussion seem contrived, excessive, and may ignore other conclusions in the literature.  The Discussion could be significantly shortened as much of it is not really relevant to the nascent data in the paper.

Ultra-conserved noncoding elements are a very interesting phenomena for which the function has yet to be fully understood.  The authors first goal is to understand what feature might constitute a biomarker that is associated with their unusual stability, and a second goal is to understand the “very strange mutational dynamics inside UCNEs.”  

RE: First goal.

Regarding the biomarker question, the dinucleotide analysis is a worthwhile contribution as are the other analyses described in the Results, for example the lack of apparent sequence motifs that could reveal a sequence of interest.  The interpretation of the GC, CC, and GG dinucleotide biases in terms of a unique DNA conformation, however, is not supported.  While regions of DNA with certain sequence attributes or symmetry elements can adopt different conformations, the assertion that UCNEs adopt a helix with unique properties, is likely not supported by the bioinformatics analysis.  Helical structure cannot be easily construed by dinucleotide frequencies, but rather helical structures are defined over longer regions of DNA, dictated by the DNA sequence.  The authors would need to analyze longer sequence regions to ascertain if a helix with unique structure were recurrent in these UCNE tracts.  However, it is noted by these authors and others that each UCNE has a distinct sequence.  While a GC bias may be common, distinct sequences would likely dictate different shapes of the 200 bp helical regions.  An unusual and unique helical structure common to UCNEs based on a GC, CC, GG biases in the collection of unique UCNE tracts seems unlikely.   It is a hypothesis that would need testing.

The lengthy discussion of base stacking stabilities and dynamics presented in Section 4.2, seems unwarranted given that the authors state, “Because of current inconsistencies, we are unable to make precise calculations of stacking inside UCNEs vs. genomic sequences and to draw final conclusions on 3D structure of UCNEs.”  The lengthy discussing is simply not supportive of the suggestion of the existence of a unique DNA conformation.  While an unusual or unique helix might bind specific proteins, DNA protein interaction requires specific interactions between amino acids and functional groups on nucleotides in the major or minor grooves and often interactions with phosphates.  Specific proteins binding to UCNEs may recognize helix shape, but specific interaction with GC dinucleotides, unless they were distributed at specific repeating intervals in all unique-sequence UCNEs seems unlikely.  The McCole (2018) observation of the unique 3D organization in genomes of UCNEs may be a result of specific protein binding, but it also may be the result of the lack of any protein, or specific protein, binding.  What is known about the organization of these sequences into nucleosomes?  If that is known, it may reveal something about the physical nature of DNAs comprising the UCNEs.

RE: second goal (section 4.3).           

Explaining the very strange mutational dynamics inside UCNEs in this paper has issues.  

For one, the idea that these sequences remain remarkably stable is well known and I question if the length of the discussion of such in this manuscript is really necessary.  It seems this discussion could be shortened.  

Can the authors define what they mean by “malignant mutation” in line 460, and describe how it is relevant to the extant results?

After several readings, I am a bit uncertain as to how “Figure 1 shows that an unknown process effectively prevents numerous novel (rare) mutations within UCNEs to spread within the population” (lines 467-469).  Perhaps this could be explained a bit more?  

The authors argue that mutations are found in UCNEs but that there is no fixation of mutation (by which I assume means the sequence of individual UCNEs resist genetic change over time).  A mutation that renders the UCNE non functional would clearly be selected against.  There are diseases associated with mutation in UCNEs (Habic et al., 2019) and deletions of UCNEs within transcriptional enhancers in mice can have deleterious effects (see discussion and references in Habic).  Therefore, mutations that change some function could certainly contribute to strong selection.  Gamete fitness could be one explanation although it does not appear that mice with deletions show reduced gamete fitness.

I do not know what to make of the conjecture of some unknown DNA repair process for these sequences.   When and where would it operate?  This seems unlikely.  

Reviewer 3 Report

The authors for the first time found a clear distinction in nucleotide composition between ultra-conserved human sequences and the entire genome. In addition, they provided evidence that abnormalities in dinucleotide composition may be connected to the DNA conformation of ultra-conserved sequences.  These are important observations that makes this paper interesting for a general reader.  

The authors hypothesized that extraordinary conservation of primary sequences of UCNE elements may be caused by peculiarities in staking interactions of adjacent nucleotides. They discussed major uncertainties and controversies in modern interpretations of nucleotide stacking.  To make this topic more understandable for a reader, I recommend making additional reference to a  thorough review by Sponer J, Sponer JE, Mládek A, Jurečka P, Banáš P, Otyepka M. entitled “Nature and magnitude of aromatic base stacking in DNA and RNA: Quantum chemistry, molecular mechanics, and experiment” (2013). 

I had a trouble in finding supplementary materials for this paper on the GENES website, but the link to the laboratory web page http://bpg.utoledo.edu/~afedorov/lab/UCNE.html provided at the end of the paper worked well. It contains all listed supplementary materials.  I found Supplementary figures and table useful enough, but the legends for the Figures S1, S2 and Tables S1, S2 must be expanded.

Table 3 in the paper is too lengthy.  It must be reformatted from one-sided to two-sided and should be fitted on a single page of publication.

It should be said that the methods of statistical data analysis are not completely clear, I would like to see more reasonable conclusions in this part. 

The language is generally good and understandable. There are several spelling mistakes, like the word  “researchers” at the very end of the paper. The authors must accomplish careful spellchecking of this paper before publication.

There is no “conclusions” section at the end of the paper.  I recommend writing a concise list of major conclusions.

Round 2

Reviewer 2 Report

The revisions made by the authors significantly improve the manuscript.